# Improving Composition of Sentence Embeddings through the Lens of Statistical Relational Learning

## Abstract

Various NLP problems – such as the prediction of sentence similarity, entailment, and discourse relations – are all instances of the same general task: the modeling of semantic relations between a pair of textual elements. We call them textual relational problems. A popular model for textual relational problems is to embed sentences into fixed size vectors and use composition functions (e.g. difference or concatenation) of those vectors as features for the prediction. Meanwhile, composition of embeddings has been a main focus within the field of Statistical Relational Learning (SRL) whose goal is to predict relations between entities (typically from knowledge base triples). In this work, we show that textual relational models implicitly use compositions from baseline SRL models. We show that such compositions are not expressive enough for several tasks (e.g. natural language inference). We build on recent SRL models to address textual relational problems, showing that they are more expressive, and can alleviate issues from simpler compositions. The resulting models significantly improve the state of the art in both transferable sentence representation learning and relation prediction.

## 1 Introduction

Predicting relations between textual units is a widespread task, essential for discourse analysis, dialog systems, information retrieval, or paraphrase detection. Since relation prediction often requires a form of understanding, it can also be used as a proxy to learn transferable sentence representations. Several tasks that are useful to build sentence representations are derived directly from text structure, without human annotation: sentence order prediction (Logeswaran et al., 2016; Jernite et al., 2017), the prediction of previous and subsequent sentences (Kiros et al., 2015; Jernite et al., 2017), or the prediction of explicit discourse markers between sentence pairs (Nie et al., 2017; Jernite et al., 2017). Human labeled relations between sentences can also be used for that purpose, e.g. inferential relations (Conneau et al., 2017). While most work on sentence similarity estimation, entailment detection, answer selection, or discourse relation prediction seemingly uses task-specific models, they all involve predicting whether a relation $R$ holds between two sentences $s_1$ and $s_2$. This genericity has been noticed in the literature before (Baudiš et al., 2016) and it has been leveraged for the evaluation of sentence embeddings within the SentEval framework (Conneau et al., 2017).

A straightforward way to predict the probability of $(s_1, R, s_2)$ being true is to represent $s_1$ and $s_2$ with $d$-dimensional embeddings $h_1$ and $h_2$, and to compute sentence pair features $f(h_1, h_2)$, where $f$ is a composition function (e.g. concatenation, product, ...). A softmax classifier $g_\theta$ can learn to predict $R$ with those features. $g_\theta \circ f$ can be seen as a reasoning based on the content of $h_1$ and $h_2$ (Socher et al., 2013). We address here limitations of existing compositions for textual relational learning.

Our contributions are as follows:

- – we review composition functions used in textual relational learning and show that they lack expressiveness (section 2);
- – we draw analogies with existing SRL models (section 3) and design new compositions inspired from SRL (section 4);

     – we perform extensive experiments to test composition functions and show that some of
       them can improve the learning of representations and their downstream uses (section 6).

## 2   COMPOSITION FUNCTIONS FOR RELATION PREDICTION

We review here popular composition functions used for relation prediction based on sentence embeddings. Ideally, they should simultaneously fulfill the following minimal requirements:

     – make use of interactions between representations of sentences to relate;

     – allow for the learning of asymmetric relations (e.g. entailment, order);

     – be usable with high dimensionalities ($\theta$ and $f$ parameters should fit in GPU memory, e.g.
       12GB).

If the main goal is transferable sentence representation learning, compositions should also incentivize sentences with a monotonic attributes change (e.g. sentences subjectivity increasing) to lie on a linear manifold, since transfer usually uses linear models. This use case is the main focus of this paper. Another goal can be learning of transferable relation representation. Concretely, a sentence encoder and $f$ can be trained on a base task, and $f(h_1, h_2)$ can be used as features for transfer in another task. In that case, the geometry of the sentence embedding space is less relevant, as long as $f(h_1, h_2)$ space works well for transfer learning. This case will also occur in our evaluations.

A straightforward instantiation of $f$ is concatenation (Hooda & Kosseim, 2017):

$$f_{[,]}(h_1, h_2) = [h_1, h_2] \tag{1}$$

However, interactions between $s_1$ and $s_2$ cannot be modeled with $f_{[,]}$ followed by a softmax regression. Using a multi-layer perceptron before the softmax would solve this issue, but it harms sentence representation learning (Conneau et al., 2017; Logeswaran & Lee, 2018), possibly because the perceptron allows for accurate predictions even if the sentence embeddings lie in a convoluted space. Consider a paraphrase detection task: given a sentence $s_1$, the sentence $s_2$ maximizing the probability of $s_2$ being an $s_1$ paraphrase does not even depend on $s_1$ with this composition. This effect has been noticed in Levy et al. (2015) regarding lexical relations. To promote interactions between $h_1$ and $h_2$, element-wise product has been used in Baudiš et al. (2016):

$$f_\odot(h_1, h_2) = h_1 \odot h_2 \tag{2}$$

Absolute difference is another solution for sentence similarity (Mueller & Thyagarajan, 2016), and its element-wise variation may equally be used to compute informative features:

$$f_-(h_1, h_2) = |h_1 - h_2| \tag{3}$$

The latter two were combined into a popular instantiation (Tai et al., 2015; Kiros et al., 2015; Mou et al., 2015):

$$f_{\odot-}(h_1, h_2) = [h_1 \odot h_2, |h_2 - h_1|] \tag{4}$$

Although effective for certain similarity tasks, $f_{\odot-}$ is symmetrical, and should be a poor choice for tasks like entailment prediction or prediction of discourse relations. For instance, if $R_e$ denotes entailment and $(s_1, s_2)$= ("It just rained", "The ground is wet"), $(s_1, R_e, s_2)$ should hold but not $(s_2, R_e, s_1)$. The $f_{\odot-}$ composition function is nonetheless used to train/evaluate models on entailment (Conneau et al., 2017) or discourse relation prediction (Nie et al., 2017).

Sometimes $[h_1, h_2]$ is concatenated to $f_{\odot-}(h_1, h_2)$ (Ampomah et al., 2016; Nie et al., 2017; Conneau et al., 2017; Shen et al., 2017). While the resulting composition is asymmetrical, that asymmetrical component involves no interaction as noted previously.

An outer product $\otimes$ has been used instead for asymmetric multiplicative interaction (Jernite et al., 2017):

$$f_\otimes(h_1, h_2) = h_1 \otimes h_2 \text{ where } (h_1 \otimes h_2)_{i,j} = h_{1i}h_{2j} \tag{5}$$

This formulation is expressive but it forces $g_\theta$ to have $d^2$ parameters per relation, which is prohibitive when there are many relations and $d$ is high. It can also hinder sentence representation learning. The problems outlined above are well known in SRL.

To sum up, existing compositions (except $f_\otimes$) can only model relations superficially for tasks currently used to train state of the art sentence encoders, like NLI or discourse connectives prediction.

| Model | Scoring function | Relation parameters |
|---|---|---|
| Unstructured (Bordes et al., 2013a) | $\|e_1 - e_2\|_p$ | - |
| TransE (Bordes et al., 2013b) | $\|e_1 + w_r - e_2\|_p$ | $w_r \in \mathbb{R}^d$ |
| RESCAL (Nickel et al., 2011) | $e_1^T W_r e_2$ | $W_r \in \mathbb{R}^{d^2}$ |
| DistMult (Yang et al., 2015) | $< e_1, w_r, e_2 >$ | $w_r \in \mathbb{R}^d$ |
| ComplEx (Trouillon et al., 2016) | $\text{Re} < e_1, w_r, \overline{e_2} >$ | $w_r \in \mathbb{C}^d$ |

Table 1

Selected relational learning models. Following Trouillon et al. (2016), $< a, b, c >$ denotes $\sum_k a_k b_k c_k$. $\text{Re}(x)$ is the real part of $x$, and $p$ is commonly set to 1.

## 3 STATISTICAL RELATIONAL LEARNING MODELS

In this section we introduce the context of statistical relational learning (SRL) and relevant models. Recently, SRL has focused on efficient and expressive relation prediction based on embeddings.

A core goal of SRL (Getoor & Taskar, 2007) is to induce whether a relation $R$ holds between two arbitrary entities $e_1, e_2$. As an example, we would like to assign a score to $(e_1, R, e_2)$ = (Paris, LOCATED_IN, France) that reflects a high probability. In embedding-based SRL models, entities $e_i$ have vector representations in $\mathbb{R}^d$ and a scoring function reflects truth values of relations. The scoring function should allow for relation-dependent reasoning over the latent space of entities. Scoring functions can have relation-specific parameters, which can be interpreted as relation embeddings. Table 1 presents an overview of a number of state of the art relational models. We can distinguish two families of models: subtractive and multiplicative.

The TransE scoring function is motivated by the idea that translations in latent space can model analogical reasoning and hierarchical relationships. Dense word embeddings trained on tasks related to the distributional hypothesis naturally allow for analogical reasoning with translations without explicit training on this task (Mikolov et al., 2013). TransE can be seen as a generalization of the older Unstructured model. We call them subtractive models.

The RESCAL, Distmult, and ComplEx scoring functions can be seen as dot product matching between $e_1$ and a relation-specific linear transformation of $e_2$ (Liu et al., 2017). This transformation helps checking whether $e_1$ matches with some aspects of $e_2$. This is well suited for similarity learning or to check associations of characteristics between entities. RESCAL allows a full linear mapping $W_r e_2$ but has a high complexity, while Distmult is restricted to a component-wise weighting $w_r \odot e_2$. ComplEx has fewer parameters than RESCAL but still allows for the modeling of asymmetrical relations. It can be interpreted as a variation of Distmult using a Hermitian product on complex embeddings instead of a dot product on real embeddings. As shown in Liu et al. (2017), ComplEx boils down to a restriction of RESCAL where $W_r$ is a block diagonal matrix. These blocks are 2-dimensional, antisymmetric and have equal diagonal terms. Using such a form, even and odd indexes of $e$'s dimensions play the roles of real and imaginary numbers respectively. The ComplEx model (Trouillon et al., 2016) and its variations (Lacroix et al., 2018) yields state of the art performance on knowledge base completion on numerous evaluations.

## 4 TEXTUAL RELATIONAL LEARNING COMPOSITION FUNCTIONS AS SRL MODELS

We call "textual relational models" relational learning models where sentence embeddings $h_i$ act as entity embeddings $e_i$, as shown in figure 1. In the following we focus on sentence embeddings, although the model can be straightforwardly applied to other levels of language granularity (such as words, clauses, or documents).

Some models (Chen et al., 2017b; Seo et al., 2016; Gong et al., 2018) do not rely on explicit sentence encodings to perform relation prediction. They combine information of input sentences at earlier stages, using conditional encoding or cross-attention. There is however no straightforward way to derive transferable sentence representations in this setting, and so these models are out of the scope

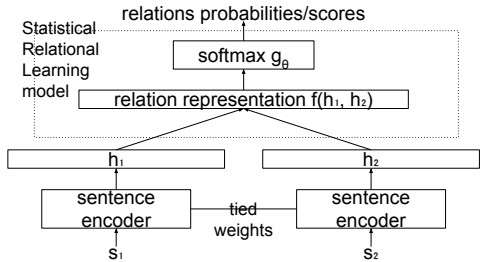

Figure 1: Overview of our relational model

of this paper. They sometimes make use of composition functions, so our work could still be relevant to them in some respect.

In this section we will make a link between sentence composition functions and SRL scoring functions, and propose new scoring functions drawing inspiration from SRL.

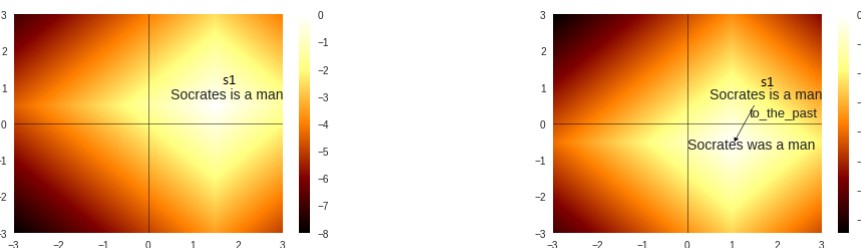

(a) Score map of $(s_1, R_{to\_the\_past}, s_2)$ over possible sentences $s_2$ using Unstructured composition.

(b) Score map of $(s_1, R_{to\_the\_past}, s_2)$ over possible sentences $s_2$ using TransE composition.

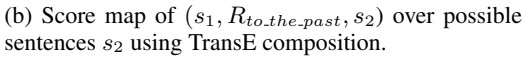
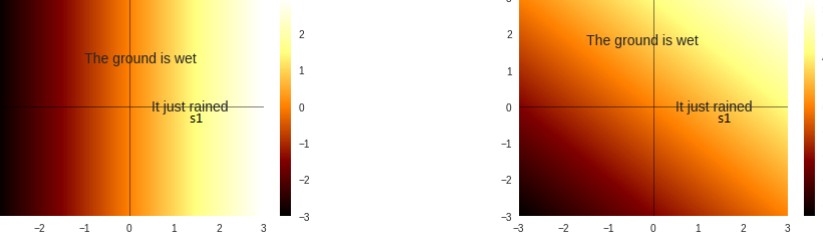

(c) Score map of $(s_1, R_{entailment}, s_2)$ over possible sentences $s_2$ using DistMult composition.

(d) Score map of $(s_1, R_{entailment}, s_2)$ over possible sentences $s_2$ using ComplEx composition.

Figure 2: Scoring function values according to different composition functions. $s_1$ and $R$ are fixed and color brightness reflects likelihood of $(s_1, R, s_2)$ for each position of $s_2$ embedding. (b) and (d) are respectively more expressive than (a) and (c).

## 4.1 LINKING COMPOSITION FUNCTIONS AND SRL MODELS

The composition function $f_\odot$ from equation 2 followed by a softmax regression yields a score whose analytical form is identical to the Distmult model score described in section 3. If $\theta_R$ denotes the softmax weights for relation $R$, the logit score for the truth of $(s1, R, s2)$ is $f(h_1, h_2)\theta_R = (h_1 \odot h_2)\theta_R$ which is equal to the Distmult scoring function $< h_1, \theta_R, h_2 >$ if $h$ acts as entity embedding and $\theta_R$ as the relation weight $w_R$ .

Similarly, the composition $f_-$ from equation 3 followed by a softmax regression can be seen as an element-wise weighted score of Unstructured (both are equal if softmax weights are all unitary).

Thus, $f_{\odot-}$ from 4 (with softmax regression) can be seen as a weighted ensemble of Unstructured and Distmult. These two models are respectively outperformed by TransE and ComplEx on knowledge base link prediction by a large margin (Trouillon et al., 2016; Bordes et al., 2013a). We therefore

propose to change the Unstructured and Distmult in $f_{\odot -}$ such that they match their respective state of the art variations in the following sections. We will also show the implications of these refinements.

## 4.2 CASTING TRANSE AS A COMPOSITION

Simply replacing $|h_2 - h_1|$ with

$$f_t(h_1, h_2) = |h_2 - h_1 + t|, \text{ where } t \in \mathbb{R}^d \tag{6}$$

would make the model analogous to TransE. $t$ is learned and is shared by all relations. A relation-specific translation $t_R$ could be used but it would make $f$ relation-specific and disallow the use of a standard softmax. Instead, here, each dimension of $f_t(h_1, h_2)$ can be weighted according to a given relation. Non-zero $t$ makes $f_t$ asymmetrical and also yields features that allow for the checking of an analogy between $s_1$ and $s_2$. Sentence embeddings often rely on pre-trained word embeddings which have demonstrated strong capabilities for analogical reasoning. Some analogies, such as *part-whole*, are computable with off-the-shelf word embeddings (Chen et al., 2017a) and should be very informative for natural language inference tasks with sentence embeddings. As an illustration, let us consider an artificial semantic space (depicted in figures 2a and 2b) where we posit that there is a "to the past" translation $t$ so that $h_1 + t$ is the embedding of a sentence $s_1$ changed to the past tense. Unstructured is not able to leverage this semantic space to correctly score $(s_1, R_{to\_the\_past}, s_2)$ while TransE is well tailored to provide highest scores for sentences near $h_1 + \hat{t}$ where $\hat{t}$ is an estimation of $t$ that could be learned from examples.

## 4.3 CASTING COMPLEX AS A COMPOSITION

Let us partition $h$ dimensions into two equally sized groups $\mathcal{R}$ and $\mathcal{I}$. For instance, they could be even and odd dimension indices of $h$. We propose a new function $f_{\mathbb{C}}$ as a way to fit the ComplEx scoring function into a composition function.

$$f_{\mathbb{C}}(h_1, h_2) = [h_1^{\mathcal{R}} \odot h_2^{\mathcal{R}} + h_1^{\mathcal{I}} \odot h_2^{\mathcal{I}}, h_1^{\mathcal{R}} \odot h_2^{\mathcal{I}} - h_1^{\mathcal{I}} \odot h_2^{\mathcal{R}}] \tag{7}$$

$f_{\mathbb{C}}(h_1, h_2)$ multiplied by softmax weights $\theta_r$ is equivalent to the ComplEx scoring function $\text{Re} < h_1, \theta_r, \overline{h_2} >$. The first half of $\theta_r$ weights corresponds to the real part of ComplEx relation weights while the last half corresponds to the imaginary part.

$f_{\mathbb{C}}$ is to the ComplEx scoring function what $f_{\odot}$ is to the DistMult scoring function. Intuitively, ComplEx is a minimal way to model interactions between distinct latent dimensions while Distmult only allows for identical dimensions to interact.

Let us consider a new artificial semantic space (shown in figures 2c and 2d) with the first dimension activating when a sentence means that it just rained, and the second dimension activating when the ground is wet. Over this semantic space, Distmult is only able to recognize entailment for paraphrases whereas ComplEx is also able to naturally model that ("it just rained", $R_{entailment}$, "the ground is wet") should be high while its converse should not.

We also propose two more general versions of $f_{\mathbb{C}}$ :

$$f_{\mathbb{C}^\alpha}(h_1, h_2) = [h_1^{\mathcal{R}} \odot h_2^{\mathcal{R}}, h_1^{\mathcal{I}} \odot h_2^{\mathcal{I}}, h_1^{\mathcal{R}} \odot h_2^{\mathcal{I}} - h_1^{\mathcal{I}} \odot h_2^{\mathcal{R}}] \tag{8}$$

$$f_{\mathbb{C}^\beta}(h_1, h_2) = [h_1^{\mathcal{R}} \odot h_2^{\mathcal{R}}, h_1^{\mathcal{I}} \odot h_2^{\mathcal{I}}, h_1^{\mathcal{R}} \odot h_2^{\mathcal{I}}, h_1^{\mathcal{I}} \odot h_2^{\mathcal{R}}] \tag{9}$$

$f_{\mathbb{C}^\alpha}$ can be seen as Distmult concatenated with the asymmetrical part of ComplEx and $f_{\mathbb{C}^\beta}$ can be seen as RESCAL with unconstrained block diagonal relation matrices. These compositions have higher dimensionality but this is not as problematic as in SRL (Freebase contains $35k$ relation types which can make it hard to learn high dimensional relation embeddings). NLP problems tend to have a moderate number of relations and we can afford to use slightly more relation parameters.

## 5 ON THE EVALUATION OF RELATIONAL MODELS

The SentEval framework (Conneau et al., 2017) provides a general evaluation for transferable sentence representations, with open source evaluation code. One only needs to specify a sentence

| name | N | task | C | representation(s) used |
|------|---|------|---|------------------------|
| MR | 11k | sentiment (movies) | 2 | $h_1$ |
| SUBJ | 10k | subjectivity/objectivity | 2 | $h_1$ |
| MPQA | 11k | opinion polarity | 2 | $h_1$ |
| TREC | 6k | question-type | 6 | $h_1$ |
| SICK$_s^m$ | 10k | NLI | 3 | $f_{m,s}(h_1, h_2)$ |
| MRPC$_s^m$ | 4k | paraphrase detection | 2 | $(f_{m,s}(h_1, h_2) + (f_{m,s}(h_2, h_1))/2$ |
| PDTB$_s^m$ | 17k | discursive relation | 5 | $f_{m,s}(h_1, h_2)$ |
| STS14 | 4.5k | similarity | - | $\cos(h_1, h_2)$ |

Table 2: Transfer evaluation tasks. N = number of training examples; C = number of classes if applicable. $h_1, h_2$ are sentence representations, $f_{m,s}$ a composition function from section 4.

encoder function, and the framework performs classification tasks or relation prediction tasks using cross-validated logistic regression on embeddings or composed sentence embeddings. Tasks include sentiment analysis, entailment, textual similarity, textual relatedness, and paraphrase detection. These tasks are a rich way to train or evaluate sentence representations since in a triple $(s_1, R, s_2)$, we can see $(R, s_2)$ as a label for $s_1$ (Baudiš et al., 2016). Unfortunately, the relational tasks hard-code the composition function from equation 4. From our previous analysis, we believe this composition function favors the use of contextual/lexical similarity rather than high-level reasoning and can penalize representations based on more semantic aspects. This bias could harm research since semantic representation is an important next step for sentence embedding. Training/evaluation datasets are also arguably flawed with respect to relational aspects since several recent studies (Dasgupta et al., 2018; Poliak et al., 2018; Levy et al., 2018; Glockner et al., 2018) show that InferSent, despite being state of the art on SentEval evaluation tasks, has poor performance when dealing with asymmetrical tasks and non-additive composition of words. In addition to providing new ways of training sentence encoders, we will also extend the SentEval evaluation framework with a more expressive composition function when dealing with relational transfer tasks, which improves results even when the sentence encoder was not trained with it.

## 6 EXPERIMENTS

Our goal is to show that transferable sentence representation learning and relation prediction tasks can be improved when our expressive compositions are used instead of the composition from equation 4. We train our relational model adaptations on two relation prediction base tasks ($\mathcal{T}$), one supervised ($\mathcal{T} = NLI$) and one unsupervised ($\mathcal{T} = Disc$) described below, and evaluate sentence/relation representations on base and transfer tasks using the SentEval framework in order to quantify the generalization capabilities of our models. The code for our experiments will be publicly available.

### 6.1 TRAINING TASKS

Natural language inference ($\mathcal{T} =$ NLI)'s goal is to predict whether the relation between two sentences (premise and hypothesis) is *Entailment*, *Contradiction* or *Neutral*. We use the SNLI dataset (Bowman et al., 2015) which contains $570k$ examples based on Flickr captions and manual annotations. We also use the MNLI dataset (Williams et al., 2017) which amounts to $433k$ examples across various domains. We call the resulting dataset AllNLI. Conneau et al. (2017) claim that NLI data allows universal sentence representation learning. They used the $f_{\odot,-}$ composition function with concatenated sentence representations in order to train their *Infersent* model.

We also train on the prediction of discourse connectives between sentences/clauses ($\mathcal{T} =$ Disc). Discourse connectives make discourse relations between sentences explicit. In the sentence *I live in Paris but I'm often elsewhere*, the word *but* highlights that there is a contrast between the two clauses it connects. We use Malmi et al.'s (2017) dataset of selected $400k$ instances with 20 discourse connectives (e.g. *however*, *for example*) with the provided train/dev/test split. This dataset has no other supervision than the list of 20 connectives. Nie et al. (2017) used $f_{\odot,-}$ concatenated with the sum of sentence representations to train their model, *DisSent*, on a similar task and showed that their encoder was general enough to perform well on SentEval tasks. They use a not yet available dataset.

| m,s | MR | SUBJ | MPQA | TREC | MRPC$^{\odot}_{-}$ | PDTB$^{\odot}_{-}$ | SICK$^{\odot}_{-}$ | STS14 | $\mathcal{T}$ |
|------|------|------|------|------|------|------|------|------|------|
| | | | Models trained on natural language inference ($\mathcal{T} = NLI$) | | | | | | |
| $\odot, -$ | 81.2 | 92.7 | 90.4 | 89.6 | 76.1 | 46.7 | 86.6 | 69.5 | 84.2 |
| $\alpha, -$ | **81.4** | **92.8** | 90.5 | 89.6 | 75.4 | 46.6 | 86.7 | 69.5 | 84.3 |
| $\beta, -$ | 81.2 | 92.6 | 90.5 | 89.6 | 76.0 | 46.5 | 86.6 | 69.5 | 84.2 |
| $\odot, t$ | 81.1 | 92.7 | 90.5 | **89.7** | **76.5** | 46.4 | 86.5 | **70.0** | **84.8** |
| $\alpha, t$ | 81.3 | 92.6 | **90.6** | 89.2 | 76.2 | 47.2 | 86.5 | **70.0** | 84.6 |
| $\beta, t$ | 81.2 | 92.7 | 90.4 | 88.5 | 75.8 | **47.3** | **86.8** | 69.8 | 84.2 |

Table 3: SentEval and base task evaluation results for the models trained on natural language inference ($\mathcal{T} = NLI$); AllNLI is used for training. All scores are accuracy percentages, except STS14, which is Pearson correlation percentage.

## 6.2 EVALUATION TASKS

Table 2 provides an overview of different transfer tasks that will be used for evaluation. We added another relation prediction task, the PDTB coarse-grained implicit discourse relation task, to Sent-Eval. This task involves predicting a discursive link between two sentences among {Comparison, Contingency, Entity based coherence, Expansion, Temporal}. We followed the setup of Pitler et al. (2009), without sampling negative examples in training. MRPC, PDTB and SICK will be tested with two composition functions: besides SentEval composition $f_{\odot,-}$, we will use $f_{\mathcal{C}^{\beta},-}$ for transfer learning evaluation, since it has the most general multiplicative interaction and it does not penalize models that do not learn a translation. For all tasks except STS14, a cross-validated logistic regression is used on the sentence or relation representation. The evaluation of the STS14 task relies on Pearson or Spearman correlation between cosine similarity and the target. We force the composition function to be symmetrical on the MRPC task since paraphrase detection should be invariant to permutation of input sentences.

## 6.3 SETUP

We want to compare the different instances of $f$. We follow the setup of Infersent (Conneau et al., 2017): we learn to encode sentences into $h$ with a bi-directional LSTM using element-wise max pooling over time. The dimension size of $h$ is 4096. Word embeddings are fixed GloVe with 300 dimensions, trained on Common Crawl 840B[1]. Optimization is done with SGD and decreasing learning rate until convergence.

The only difference with regard to Infersent is the composition. Sentences are composed with six different compositions for training according to the following template:

$$f_{m,s,1,2}(h_1, h_2) = [f_m(h_1, h_2), f_s(h_1, h_2), h_1, h_2] \tag{10}$$

$f_s$ (subtractive interaction) is in $\{f_-, f_t\}$, $f_m$ (multiplicative interaction) is in $\{f_{\odot}, f_{\mathbb{C}^{\alpha}}, f_{\mathbb{C}^{\beta}}\}$. We do not consider $f_{\mathbb{C}}$ since it yielded inferior results in our early experiments using NLI and SentEval development sets.

$f_{m,s,1,2}(h_1, h_2)$ is fed directly to a softmax regression. Note that Infersent uses a multi-layer perceptron before the softmax, but uses only linear activations, so $f_{\odot,-,1,2}(h_1, h_2)$ is analytically equivalent to Infersent when $\mathcal{T} = NLI$.

## 6.4 RESULTS

Having run several experiments with different initializations, the standard deviations between them do not seem to be negligible. We decided to take these into account when reporting scores, contrary to previous work (Kiros et al., 2015; Conneau et al., 2017): we average the scores of 6 distinct runs for each task and use standard deviations under normality assumption to compute significance. Table 3 shows model scores for $\mathcal{T} = NLI$, while Table 4 shows scores for $\mathcal{T} = Disc$. For comparison, Table 5 shows a number of important models from previous work. Finally, in Table 6, we present

---

[1]https://nlp.stanford.edu/projects/glove/

| | | | | Models trained on discourse connective prediction ($\mathcal{T} = Disc$) | | | | |
|---|---|---|---|---|---|---|---|---|
| m,s | MR | SUBJ | MPQA | TREC | MRPC$^{\odot}_{\_}$ | PDTB$^{\odot}_{\_}$ | SICK$^{\odot}_{\_}$ | STS14 | $\mathcal{T}$ |
| $\odot, -$ | **80.4** | 92.7 | 90.2 | 89.5 | 74.5 | 47.3 | 83.2 | 57.9 | 35.7 |
| $\alpha, -$ | **80.4** | **92.9** | 90.2 | 90.2 | 75 | **47.9** | 83.3 | 57.8 | 35.9 |
| $\beta, -$ | 80.2 | 92.8 | 90.2 | 88.4 | 74.9 | 47.5 | 82.9 | 57.7 | 35.9 |
| $\odot, t$ | 80.2 | 92.8 | 90.2 | **90.4** | 74.6 | 48.5 | 83.4 | **58.6** | **36.1** |
| $\alpha, t$ | 80.2 | **92.9** | **90.3** | 90.3 | **75.1** | 47.8 | 83.2 | 58.3 | **36.1** |
| $\beta, t$ | 80.2 | 92.8 | **90.3** | 89.7 | 74.4 | 47.9 | **83.7** | 58.2 | 35.7 |

Table 4: SentEval and base task evaluation results for the models trained on discourse connective prediction ($\mathcal{T} = Disc$). All scores are accuracy percentages, except STS14, which is Pearson correlation percentage.

| | | | | Comparison models | | | |
|---|---|---|---|---|---|---|---|
| model | MR | SUBJ | MPQA | TREC | MRPC$^{\odot}_{\_}$ | PDTB$^{\odot}_{\_}$ | SICK$^{\odot}_{\_}$ | STS14 |
| Infersent | 81.1 | 92.4 | 90.2 | 88.2 | 76.2 | - | 86.3 | 70 |
| SkipT | 76.5 | 93.6 | 87.1 | 92.2 | 73 | - | 82.3 | 29 |
| BoW | 77.2 | 91.2 | 87.9 | 83 | 72.2 | 43.9 | 78.4 | 54.6 |

Table 5: Comparison models from previous work. InferSent represents the original results from Conneau et al. (2017), SkipT is SkipThought from Kiros et al. (2015), and BoW is our re-evaluation of GloVe Bag of Words from Conneau et al. (2017).

results for sentence relation tasks that use an alternative composition function ($f_{\mathbb{C}^\beta, \_}$) instead of the standard composition function used in SentEval.

For sentence representation learning, the baseline, $f_{\odot}-$ composition already performs rather well, being on par with the InferSent scores of the original paper, as would be expected. However, macro-averaging all accuracies, it is the second worst performing model. $f_{\mathbb{C}^\alpha, t, 1, 2}$ is the best performing model, and all three best models use the translation ($s = t$). On relational transfer tasks, training with $f_{\mathbb{C}^\alpha, t, 1, 2}$ and using complex $\mathbb{C}^\beta$ for transfer (Table 6) always outperform the baseline ($f_{\odot, -, 1, 2}$ with $\odot -$ composition in Tables 3 and 4). Averaging accuracies of those transfer tasks, this result is significant for both training tasks at level $p < 0.05$ (using Bonferroni correction accounting for the 5 comparisons). To our knowledge, it outperforms all previously reported scores (Conneau et al., 2017) on SICK Entailment task with single task training. On base tasks and the average of non-relational transfer tasks (MR, MPQA, SUBJ, TREC), our proposed compositions are on average slightly better than $f_{\odot, -, 1, 2}$. Representations learned with our proposed compositions can still be compared with simple cosine similarity: all three methods using the translational composition ($s = t$) very significantly outperform the baseline (significant at level $p < 0.01$ with Bonferroni correction) on STS14 for $\mathcal{T} = NLI$. Thus, we believe $f_{\mathbb{C}^\alpha, t, 1, 2}$ has more robust results and could be a better default choice than $f_{\odot, -, 1, 2}$ as composition for representation learning. [2]

Additionally, using $\mathbb{C}^\beta$ (Table 6) instead of $\odot$ (Tables 3 and 4) for transfer learning in relational transfer tasks (PDTB, MRPC, SICK) yields a significant improvement on average, even when $m = \odot$ was used for training ($p < 0.001$). Therefore, we believe $f_{\mathbb{C}^\beta, \_}$ is an interesting composition for inference or evaluation of models regardless of how they were trained.

# 7    RELATED WORK

There are numerous interactions between SRL and NLP. We believe that our approach merges two specific lines of work: relation prediction and modeling textual relational tasks.

Some previous NLP work focused on composition functions for relation prediction between text fragments, even though they ignored SRL and only dealt with word units. Word2vec (Mikolov et al.,

---

[2]Note that our compositions are also beneficial with regard to convergence speed: on average, each of our proposed compositions needed less epochs to converge than the baseline $f_{\odot, -, 1, 2}$, for both training tasks.

| m,s | $\mathcal{T} = NLI$ | | | $\mathcal{T} = Disc$ | | |
|---|---|---|---|---|---|---|
| | MRPC$^{\beta}_{-}$ | PDTB$^{\beta}_{-}$ | SICK$^{\beta}_{-}$ | MRPC$^{\beta}_{-}$ | PDTB$^{\beta}_{-}$ | SICK$^{\beta}_{-}$ |
| $\odot, -$ | 74.8 | 48.2 | 83.6 | **76.2** | 47.2 | 86.9 |
| $\alpha, -$ | 74.9 | **49.3** | 83.8 | 75.9 | 47.1 | 86.9 |
| $\beta, -$ | 75 | 48.8 | 83.4 | 75.8 | 47 | 87 |
| $\odot, t$ | 74.9 | 48.7 | 83.6 | **76.2** | **47.8** | 86.8 |
| $\alpha, t$ | **75.2** | 48.6 | 83.5 | **76.2** | 47.6 | **87.3** |
| $\beta, t$ | 74.6 | 48.9 | **83.9** | **76.2** | **47.8** | 87 |

Table 6: Results for sentence relation tasks using an alternative composition function ($f_{\mathbb{C}^{\beta}, -}$) during the evaluation step.

2013) has sparked a great interest for this task with word analogies in the latent space such as $\overrightarrow{king} - \overrightarrow{queen} \approx \overrightarrow{man} - \overrightarrow{woman}$. Levy & Goldberg (2014) explored different scoring functions between words, notably for analogies. Hypernymy relations were also studied, by Chang et al. (2017) and Fu et al. (2014). Levy et al. (2015) proposed tailored scoring functions. Even the skipgram model (Mikolov et al., 2013) can be formulated as finding relations between context and target words. We did not empirically explore textual relational learning at the word level, but we believe that it would fit in our framework, and could be tested in future studies. Numerous approaches (Chen et al., 2017b; Seok et al., 2016; Gong et al., 2018) were proposed to predict inference relations between sentences, but don't explicitly use sentence embeddings. Instead, they encode sentences jointly, possibly with the help of previously cited word compositions, therefore it would also be interesting to try applying our techniques within their framework.

Some modeling aspects of textual relational learning have been formally investigated by Baudiš et al. (2016). They noticed the genericity of relational problems and explored multi-task and transfer learning on relational tasks. Their work is complementary to ours since their framework unifies tasks while ours unifies composition functions. Subsequent approaches use relational tasks for training and evaluation on specific datasets (Conneau et al., 2017; Nie et al., 2017).

## 8 CONCLUSION

We have demonstrated that a number of existing models used for textual relational learning rely on composition functions that are already used in Statistical Relational Learning. By taking into account previous insights from SRL, we proposed new composition functions and demonstrated their usefulness for tasks presented in recent work. These composition functions are all simple to implement and we hope that it will become standard to try them on relational problems. Larger scale data might leverage these more expressive compositions, as well as more compositional, asymmetric, and arguably more realistic datasets (Dasgupta et al., 2018; Gururangan et al., 2018). Finally, our compositions can also be helpful to improve interpretability of embeddings, since they can help measure relation prediction asymmetry. Analogies through translations helped interpreting word embeddings, and perhaps anlyzing our learned $t$ translation could help interpreting sentence embeddings.

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
