# OpenReview forum: "Improving Composition of Sentence Embeddings through the Lens of Statistical Relational Learning"
_ICLR.cc/2019/Conference_

### Official Review · AnonReviewer3 · 2018-11-01
**What does the SRL lens contribute?**

**Rating:** 6
**Confidence:** 4

**Review:**

This paper presents a view of sentence-level prediction tasks as statistical relation learning problems. In particular the paper argues that composition functions used in recent SRL techniques developed for entity-to-entity relationship detection can be applied to sentence-level relation prediction tasks.

Suppose there is a prediction training task defined over pairs of sentences (x1, x2). This task requires some function 'f' that composes the sentence representations h1 and h2 into a single representation which is then used to
make the relation prediction i.e., we have a model g(f(h1, h2)) that is used to predict some relation between R(x1, x2).  This paper aims to show that with a better 'f' we can hope for a better result in transfer tasks (in addition to doing better on the training task).

The paper argues that this setting, at a high level, is similar to the composition function used in entity-entity relation prediction. There have been many such methods in the recent past (e.g., TransE, ComplEx, RESCAL). This paper asks whether these composition functions can work well for sentence-level tasks.

The paper then presents experiments which compare the performance of different composition functions against a basic composition function used in InferSent.

Strengths of the paper:

1. I like the main question of what can we learn from SRL. This seeks to bridge some independent research threads.
2. The evaluation considers a range of composition functions used in SRL and applies them to the sentence tasks.
3. Points out that some of the composition functions used in existing models are not particularly strong.

Issues:

I like the starting point for this paper very much and agree that the existing composition functions for sentence relations are rather weak. However, I am struggling to see if there is (i) a convincing conceptual argument for why SRL view of compositions is necessarily the answer for sentence level tasks, or (ii) a convincing empirical case for the same.  Some details on these points:

1) The parallels between entity-entity relations and sentence-sentence relations seems a bit of a stretch to me. There is always some level of abstraction at which two problems might look similar, which can be advantageous for repurposing solutions. However, in this case I think the SRL view of the world hides the complexities in sentence-sentence relation tasks (e.g. aligning relevant pieces of information, requiring more complex composition functions to derive meaning etc.).

2) I am not sure what knowledge we are getting from an SRL view of the problem that is not already known already to the communities that work on sentence embedding. The minimum requirements laid out can be met easily by existing methods for sentence representations. For instance that we need to allow for asymmetric relations (entailment order) is very well known. As the authors themselves point out there are solutions for this problem.

3) The empirical results don't appear convincing. The average gain for any particular method over InferSent is 0.3 in macro average. There is no single SRL based composition method that works consistently clear gains across most tasks.

Here are some suggestions that I think will improve the paper (or at least help me buy the motivation):

1. One question that might be useful to make a conceptual argument is how much work should be done in 'f' and should it change for the different type of target tasks.

If the idea is to transfer h for single sentence target task, then a powerful 'f' can render h1 and h2 to be simple enough, such that bulk of the work in extracting task related information might be done by 'f' itself. Therefore, transferred h may not be as powerful as it could have been with a less powerful 'f'.

If the idea is to transfer f(h1, h2) for sentence-pair target tasks, then a powerful 'f' might be a good thing.

2) Another useful discussion would be to discuss why more powerful alignment based sentence representations are not being considered at least for comparison purposes.

The paper wants to go from a simple 'f' (i.e. concat(h1,h2), h1-h2) to some other choices for 'f' that are known functions from SRL.

There are several sentence-level representation functions such as ESIM [Chen et al., 2016] which uses a combined representation of premise and hypothesis sentences using soft alignment to specifically address the issues in comparing sentences. A similar representation is computed in BiDAF [Seo et al., 2017] in the context of matching question representation with sentence representations.

To summarize, I really like the basic starting point for the paper and would love to see a more compelling presentation of the conceptual argument and a stronger empirical comparison.

---

> ### Public Comment · (anonymous) · 2018-11-12
> **Response to reviewer #3**
>
> Many thanks for your helpful comments and constructive criticism. See also our general answers that discuss some of the points you make here (especially point 3 and suggestion 2).
> As a complement:
> Complexity (word compositions, alignment of relevant informations) can be indeed emphasized during either sentence encoding, either in the composition of sentence encodings.
>
> It makes sense to delegate the complexity in sentence encoding for transferable sentence representations (transfer from h representation, e.g. SUBJ, MR tasks).
> However insufficient complexity can be detrimental to sentence embeddings if the compositions are so simple that sentence embeddings can’t leverage base task training data properly. In the case of natural language inference, if a dot product composition is used, the model will be insensitive to premise/hypothesis order.
> Conversely, if the complexity is too high (e.g. deep multilayer perceptron with [h1,h2] as input), sentences with a monotonic characteristic change (e.g. sentences generality increasing) have low incentive to lie on a linear manifold, so transfer learning performance (with linear predictors) can be lowered
> This use case has been the main focus of the paper.
>
> Composed representations (i.e. f(h1,h2) ) can be also be used for transfer in relation prediction tasks. In that case, assuming the base training task incentivizes good composed representations, it makes sense to make f complex and possibly non-linear with respect to h1 and h2
> This aspect have been explored in the paper, too, since we use custom composition on relational tasks (e.g. SICK PDTB) sometimes the same as composition used with the base task training. But further investigations could be done regarding this topic. Attention mechanisms and non-linear functions could be used and transferred. But the compositions we proposed could still be provide useful features for them.

---

### Official Review · AnonReviewer1 · 2018-11-02
**A new learning representation for compositions of text embeddings is proposed and evaluated on two NLP tasks.**

**Rating:** 5
**Confidence:** 3

**Review:**

The paper describes an new representation for compositions of text embeddings using ideas from statistical relational learning.

The work is based on the premise that existing simple compositions are not very effective for relational problems in NLP such as semantic similarity, entailment etc. Therefore, the authors propose to use more complex compositions of embeddings to learn richer representations that can be useful several NLP problems that need to relate embeddings. The main idea is to develop compositions for language models based on SRL methods that learn relationships between entities such as ReSCAL and IETrans.

Compositions based on deep semantic meaning in language is a significant problem. The proposed ideas seem to be quite general for NLP compositions. However, some of the listed contributions were not so clear to me. For example, in section 2, it seems like the results were already known that some of the compositions are weak (or maybe the way it is written needs to be changed a little). Regarding the novelty, w.r.t the compositions, it does use existing SRL work but in a different context of NLP problems, this makes novelty a bit weak.

Regarding the experiments, several NLP datasets are used for evaluation across 2 tasks, showing that the method can generalize well.  However, the improvement over existing methods is marginal. Are there tradeoffs w.r.t training time etc. since the compositions are more complex? There is a sentence about it but a little vague.

---

> ### Public Comment · (anonymous) · 2018-11-12
> **Response to reviewer #2**
>
> Many thanks for your helpful comments and constructive criticism.
> -See the general answer regarding the novelty and magnitude of improvements.
> - The baseline composition needed 15.3 epochs on average vs 14 for other compositions on NLI task. On Disc task, the difference wasn’t statistically significant (baseline composition needed 0.1 more epoch). Additional computational cost per epoch is negligible, so our models actually converge faster.

---

### Official Review · AnonReviewer2 · 2018-11-02
**SRL composition functions for Sentence Embedding Tasks**

**Rating:** 5
**Confidence:** 3

**Review:**

This work proposes a view on tasks requiring pairs of sentence representations from the perspective of statistical relational learning (SRL), where there exist a multitude of composition functions for pairs of entity vectors. The authors systematically categorise different types of composition functions and apply them in tasks for testing sentence representations, with two sentence representation pretraining tasks.

Strengths:
- proposition of a unifying view onto two mostly separate strands of research
- principled way of thinking about the requirements to a relational vector composition model
- systematicity - several composition functions are categorised and systematically compared
- breadth: comparison of a large variety of composition functions on multiple tasks with two separate sentence embedding pretraining tasks
- good practice: significance testing of results.

Weaknesses:
- empirical differences are marginal, but authors claim to “significantly improve the state of the art […]” in the abstract, which I do not see as justified. The major source of variation appears to be the pretraining task chosen, not the composition function.
- The chosen scope is limited: here sentence representations cannot depend on one another, whereas this is commonplace in practice, e.g. via per-token attention on the other sequence, or in conditional LSTMs.
- the observations on expressiveness of composition functions in SRL are not new. It would have been interesting to see particular ways in which these observations differ from SRL when lifted over to the new context of sentence representations, rather than entity pairs.
- The proposed method boils down to combining relatively simple components in a straightforward manner, little innovation in terms of methodology.
- related work mostly discusses word-level representations, whereas the paper is about sentence-level representations and SRL.
- particular formulations and claims are in several points unclear, vague, imprecise or too broad — see other comments below.


Other comments:
- imprecise: “reasoning over its input embeddings“. Can this be made more specific?
- Section 2 unclear: “additive and weak“ interaction. What does that mean?
- Section 2,  bullet point 3: imprecise, and motivation unclear. Perhaps: formulate in terms of computational complexity?
- Broad claim - be more specific: “The ComplEx model yields state of the art performance on knowledge base completion“ — other models have been proposed, many of which outperform ComplEx on specific datasets, so the claim in its full generality cannot hold.
- strong wording in abstract: “we prove that textual relational models implicitly use compositions from baseline SRL models”. In my personal view this is not strong enough a theoretical result to “prove” it.
- speculative/unclear: “expressive compositions can also be helpful to improve interpretability and evaluation of sentence embeddings by providing sentence level analogies”
- speculative and vague: “From our previous analysis, we believe this composition function favours the use of contextual/lexical similarity rather than high-level reasoning and can penalise representations based on more semantic aspects. This bias could harm research since semantic representation in an important next step for sentence embedding”
- Section 4.2: initially unclear goal of this section
- Four of the Transfer evaluation tasks only use one embedding - h1. While it is interesting to have results also on these tasks, these are somewhat unrelated to the main topic of the paper
- some more details on significance tests would be useful. Normality assumption? Number of re-runs?

---

> ### Public Comment · (anonymous) · 2018-11-12
> **Reponse to reviewer #1**
>
> Many thanks for your helpful comments and constructive criticism.
> -See the general answer regarding marginal empirical differences
> -See the general answer regarding the scope of the model.
> -Carrying over expressiveness observation from SRL to sentence relation prediction is one goal of the paper. Please note that insufficiently expressive compositions have been used in influential papers (InferSent/SentEval, [Subramanian2018])
> -We focused on unifying two fields and evaluating whether or not it could improve sentence embedding and relation prediction. Improvements are modest but statistically significant, and it could be worth investigating SRL losses or other sophistications.
> -We cite related work on textual relation prediction task with a focus on composition. They happen to mostly deal with word level tasks.
>
> Regarding other comments, we will also take them in account in the updated version of the article:
> -The relation score is predicted from a learned function based on the input information from e1 and e2 embeddings. This function can be viewed as reasoning on the basis of their content
> -By ”weak”, we mean that h1 and h2 have independent contributions to a relation score. With Hadamard product composition, changes in h1 changes how h2 contributes to a relation score.
> -There is no clear-cut criterion for usability with high dimension, however, as said on equation 5 commentary, d^2 parameters per relation is too memory demanding for most modern GPUs.
> -New results based on ComplEx model [Lacroix2018] are the highest published on many tasks, but we meant that ComplEx was generally on par with the state of the art (as opposed to distmult).
> -Section 4.1 establishes identity between formulations of sentence compositions and SRL models which haven’t been shown before, to the best of our knowledge. Showing that identity is arguably mathematically trivial, but we believe it is still interesting.
> -The goal of the paper is to show that improved compositions can improve relation prediction but also sentence representations themselves, e.g. by leveraging the asymmetry of training data, so it was important to evaluate both claims.
> -We did 6 re-runs and use normality assumption
>
>
> [Lacroix2018] Timothée Lacroix et al. Canonical Tensor Decomposition for Knowledge Base Completion
> ICM2018 https://arxiv.org/pdf/1806.07297.pdf
> [Subramanian2018] Sandeep Subramanian et al., Learning General Purpose Distributed Sentence Representations Via Large Scale Multitask Learning
> ICLR2018 https://arxiv.org/pdf/1804.00079.pdf

---

### Public Comment · (anonymous) · 2018-11-12
**Addressing general points made in the reviews**

We thank the reviewers for their detailed and constructive feedback; we will provide answers to common concerns below, and respond to specific concerns in separate replies to each reviewer.
We will also update the paper and take in account all suggestions.

* Limited scope of the approach:
Reviewers raised concerns with regard to the restricted setting of our representation learning framework, for instance compared to models that allow for interactions in early stages of the prediction process. Note that this is a conscious choice. Firstly, it makes sentence representations transferable for single-sentence classification (since it requires explicit sentence representation). Secondly the restricted setting can be beneficial for relation prediction transfer learning in the context of limited target task data (for relation prediction based on two sentence encodings, fewer parameters are required when there is no attention mechanism to train). The majority of papers working on transferable sentence representations equally assume this restricted setting. Finally, note that the cited models, with interactions between sentences at earlier stages of representation learning, also implicitly use SRL compositions (ESIM [Qian Chen2017], eq 14-15, and [Seo2017] eq 2), therefore our propositions could also be incorporated and tested within the aforementioned models.

* Insufficient empirical improvement:
The results show modest, but statistically significant improvements of the proposed compositions over the basic composition from Infersent. We wanted to show to what extent expressiveness of compositions functions has an impact representations and predictions in transfer tasks. We would like to emphasize that the differences are significant when averaging over all tasks: this is discussed in section 6.4, but does not appear in the table with the breakup by task. For clarity, we separated the results with supervised (SNLI) training and unsupervised (Disc) training, and added averages of results across single sentence classification tasks, relational tasks (with basic or more expressive composition), and a global average.

* Insufficient novelty:
Some of what is proposed here might seem obvious from the SRL point of view, but more expressive compositional approaches for sentence representations are not well explored within NLP (if at all), when the goal is to train or evaluate transferable sentence representations.
Also note that we have modified SRL models for our tasks (by untying ComplEx relation weights and using a single translation vector t, , cfr. sections 4.2/4.3)


References:
[Chen2017] Qian Chen et al., Enhanced LSTM for Natural Language Inference
ACL2017 https://arxiv.org/pdf/1609.06038.pdf

[Seo2017] Minjoon Seo et al., Bi-Directional Attention Flow For Machine Comprehension
ICLR2017 https://arxiv.org/pdf/1611.01603.pdf

---

### Meta-Review · Area_Chair1 · 2018-12-13
**Interesting framing, but limited contributions**

**Confidence:** 3
**Recommendation:** Reject

**Metareview:**

This paper offers a new angle through which to study the development of comparison functions for sentence pair classification tasks by drawing on the literature on statistical relational learning. All three reviewers seemed happy to see an attempt to unify these two closely related relation-learning problems. However, none of the reviewers were fully convinced that this attempt has yielded any substantial new knowledge: Many of the ideas that come out of this synthesis have already appeared in the sentence-pair modeling literature (in work cited in the paper under review), and the proposed new methods do not yield substantial improvements for the tasks they're tested on.

I'm happy to accept the authors' arguments that sentence-to-vector models have practical value, and I'm not placing too much weight on the reviewer's comments about the choice to use that modeling framework. I am slightly concerned that the reviewers (especially R2) observed some overly broad statements in the paper, and I urge the authors to take those comments very seriously.

I'm mostly concerned, though, about the lack of an impactful positive contribution: I'd have hoped for a paper of this kind to offer a  a method with clear empirical advantages over prior work, or else a formal result which is more clearly new, and the reviewers are not convinced that this paper makes a contribution of either kind.